# Evolution and Control of COVID-19 Epidemic in Hong Kong

**DOI:** 10.3390/v14112519

**Published:** 2022-11-14

**Authors:** Shuk-Ching Wong, Albert Ka-Wing Au, Janice Yee-Chi Lo, Pak-Leung Ho, Ivan Fan-Ngai Hung, Kelvin Kai-Wang To, Kwok-Yung Yuen, Vincent Chi-Chung Cheng

**Affiliations:** 1Infection Control Team, Queen Mary Hospital, Hong Kong West Cluster, Hong Kong SAR, China; 2Centre for Health Protection, Department of Health, Hong Kong SAR, China; 3Department of Microbiology, Li Ka Shing Faculty of Medicine, The University of Hong Kong, Hong Kong SAR, China; 4Carol Yu Center for Infection, The University of Hong Kong, Hong Kong SAR, China; 5Department of Medicine, Li Ka Shing Faculty of Medicine, The University of Hong Kong, Hong Kong SAR, China; 6Department of Microbiology, Queen Mary Hospital, Hong Kong SAR, China

**Keywords:** COVID-19, SARS-CoV-2, variants, infection control

## Abstract

Hong Kong SAR has adopted universal masking, social distancing, testing of all symptomatic and high-risk groups for isolation of confirmed cases in healthcare facilities, and quarantine of contacts as epidemiological control measures without city lockdown or border closure. These measures successfully suppressed the community transmission of pre-Omicron SARS-CoV-2 variants or lineages during the first to the fourth wave. No nosocomial SARS-CoV-2 infection was documented among healthcare workers in the first 300 days. The strategy of COVID-19 containment was adopted to provide additional time to achieve population immunity by vaccination. The near-zero COVID-19 situation for about 8 months in 2021 did not enable adequate immunization of the eligible population. A combination of factors was identified, especially population complacency associated with the low local COVID-19 activity, together with vaccine hesitancy. The importation of the highly transmissible Omicron variant kickstarted the fifth wave of COVID-19, which could no longer be controlled by our initial measures. The explosive fifth wave, which was partially contributed by vertical airborne transmission in high-rise residential buildings, resulted in over one million cases of infection. In this review, we summarize the epidemiology of COVID-19 and the infection control and public health measures against the importation and dissemination of SARS-CoV-2 until day 1000.

## 1. Introduction

Throughout the history of mankind, we have been constantly challenged by emerging infectious diseases, some of which caused devastating pandemics. Over the past 1500 years, we have experienced plague, tuberculosis, influenza, and acquired immunodeficiency syndrome [1]. The Justinian plague in the Old World and the medieval plague, caused by *Yersinia pestis* transmitted by fleas, aerosols, and contaminated food, killed half and one-third of world population, respectively [2,3]. Emerging viral infections became the predominant threat in the last century. The 1918–1919 influenza pandemic caused an estimated 50 million deaths worldwide [4]. Historical records reveal that a range of interventions was attempted to control the spread of influenza in 1918, including closure of schools and churches, banning of mass gatherings, mandated mask wearing, case isolation, and use of disinfection and hygiene measures [5]. The improvement of public health interventions and responses resulted in a marked reduction in the global number of deaths to between one and four million during the influenza pandemics in 1957 and 1968 [6,7]. With further introduction of therapeutics on top of infection control and public health measures during the influenza pandemic in 2009 [8], the estimated global number of deaths associated within the first 12 months was less than 0.3 million [9], which was apparently not particularly excessive compared with seasonal influenza viruses [10]. Moreover, the residual immunity of those above 60 years of age against the previous influenza H1 strain circulating prior to 1957 might have contributed to the relatively low mortality of the 2009 pandemic as well [11].

Coronavirus, a positive-sense, single-stranded RNA virus, is an important emerging pathogen in addition to influenza virus. Human coronavirus 229E and OC43 were consecutively identified as previously unrecognized etiological agents for the common cold in 1965 [12]. Human coronavirus NL63 and HKU-1 were subsequently discovered as respiratory pathogens in 2004 and 2005, respectively [13,14]. The occurrence of severe acute respiratory syndrome coronavirus 1 (SARS-CoV-1) in 2003 [15] and sporadic outbreaks of Middle East respiratory syndrome coronavirus since 2012 [16] in limited geographic areas had already alerted the world to the potential occurrence of devastating infectious diseases in the 21st century. In 2018, the World Health Organization (WHO) promulgated a list of priority diseases requiring urgent research and development attention [17], with the inclusion of Disease X, representing pathogens currently unknown that cause human disease and requiring cross-cutting preparedness [18,19]. Coronavirus disease 2019 (COVID-19), caused by severe acute respiratory syndrome coronavirus 2 (SARS-CoV-2), was described as the first Disease X [20]. The official announcement of the community-acquired pneumonia outbreak in Wuhan, Hubei Province, was made by the Health Commission of Hubei Province of the People’s Republic of China on 31 December 2019 (day 1) [21]. Compared with the outbreak of SARS-CoV-1 in 2003, the time to discovery of the etiological agent of this novel coronavirus, SARS-CoV-2, was shorter [22]. The genome sequences of SARS-CoV-2 obtained from infected patients were closely related to two bat-derived SARS-like coronaviruses [23,24,25,26], analogous to the finding of SARS-like coronavirus in Chinese horseshoe bats in 2005, suggesting that bats may be the ultimate origin of coronaviruses posing a threat to humans [27,28].

The WHO declared a Public Health Emergency of International Concern on 30 January 2020 and proclaimed the outbreak a pandemic on 11 March 2020 (day 72) [29]. By then, COVID-19 had already been detected in 113 countries or territories on 5 continents, resulting in 118,319 confirmed cases all over the world [30]. As of 25 September 2022 (day 1000), 612 million confirmed cases and 6.5 million deaths had been reported globally [31]. During the early phase of the COVID-19 pandemic, various forms of lockdown on a global basis [32] led to far-reaching socioeconomic and health effects in different strata of life [33]. In the cosmopolitan city of Hong Kong, where the population density was 7126 persons per square kilometer, among the top four worldwide [34], the strategy was precise and dynamic infection control and public health measures without lockdown of the city or complete border closure. We summarized our experience in the control of COVID-19 during the evolution of SARS-CoV-2 variants [35], from the wild-type virus to Omicron sub-lineages (BA.1 and BA.2 to BA.4 and BA.5), in Hong Kong.

## 2. Global Epidemiology of COVID-19

Soon after the first reported case of COVID-19, the WHO was monitoring the global epidemiology of COVID-19 by regularly updating the situation report since 21 January 2020 [36]. In addition, data on the number of COVID-19 cases and deaths was also available in the public domains of the European Centre for Disease Prevention and Control [37], academic institutions [38], and public media [39,40,41,42]. On 11 March 2022 (two years after the start of the COVID-19 pandemic), there were 447 million confirmed cases, constituting 6% of the global population, about the combined population size of the 27 countries making up the European Union, with 6 million (0.075% of the world population) deaths [43]. As of 25 September 2022 (day 1000), 14 (6.3%) of 222 countries or areas had recorded over 10 million cumulative confirmed cases (Table 1). These 14 countries accounted for 407 million cases, which constituted 67% of the COVID-19 infections in the world.

During the evolution of the COVID-19 pandemic, mutations of SARS-CoV-2 generated many variants. The WHO used letters of the Greek alphabet to define variants of concern (VOC) for better communication and discussion by the general public. VOCs are viruses associated with one or more of the following attributes at a degree of global public health significance, including (i) an increase in transmissibility or detrimental change in COVID-19 epidemiology, (ii) an increase in virulence or change in clinical disease presentation, and (iii) a decrease in effectiveness of public health and social measures or available diagnostics, vaccines, or therapeutics [35]. Currently and previously circulating VOCs are summarized in Table 2.

The emergence of SARS-CoV-2 VOCs has resulted in a progressive increase in transmissibility. The Alpha variant had a 43 to 90% higher reproduction rate than pre-existing variants [44], which caused a rapid expansion of the variant during autumn 2020 in the UK, especially among patients under 20 years of age [45]. The Delta variant was on average 63–167% more transmissible than the Alpha variant [46]. The effective reproduction number of the Omicron variant was 3.19 times greater than that of the Delta variant under the same epidemiological conditions [47]. The secondary attack rate of the Omicron variant was higher than Delta among household contacts, with a relative risk of 1.41 [48]. The evolution of SARS-CoV-2 VOCs led to a continuing rise in case numbers. The trend of COVID-19 cases in relation to the emergence of SARS-CoV-2 VOCs is illustrated in Figure 1.

## 3. Epidemiology of COVID-19 in Hong Kong

Since the outbreak of community-acquired pneumonia in Wuhan, Hubei Province, announced on 31 December 2019 (day 1), the Centre for Health Protection, Department of Health, Government of Hong Kong Special Administrative Region, China, had been closely monitoring the situation. A website was established for the public to access updated information, including epidemiological information of each laboratory-confirmed COVID-19 case [49]. The first COVID-19 case was a traveler from Wuhan who arrived in Hong Kong on 21 January 2020 (day 22), and was confirmed on 23 January 2020 (day 24), whereas the first locally acquired COVID-19 case was reported on 30 January 2020 (day 31). Within the first 1000 days, there were five waves of COVID-19 with a total of 1,745,505 cases recorded in Hong Kong (Table 3). Prior to the fifth wave, there were 12,636 confirmed COVID-19 cases in Hong Kong with an incidence of 1685 cases per million population. The median age was 43 years (range: 12 days to 100 years). The overall case fatality rate was 1.7% (213/12,636). The epidemiological information of 54 outbreaks in the first to the fourth waves has been reported previously [50]. During the fifth wave (primary Omicron wave), 1,200,068 COVID-19 cases were reported. The incidence of COVID-19 during the primary Omicron wave was 95 times higher than the total incidence from the first to the fourth wave in Hong Kong, with the daily number of new cases reaching a peak of >50,000 on 3 March 2022 (day 794). As of 31 May 2022 (day 883), there were 9318 deaths within the primary Omicron wave, of which 6632 (71.2%) and 2312 (24.8%) were aged ≥80 and 60–79 years, respectively. Among the deaths in the elderly population, 4934 (74.4%) and 1543 (66.7%) aged 80 and 60–79 years, respectively, were unvaccinated. A secondary wave of Omicron COVID-19 cases was observed when the BA.5 emerged in June 2022, which became the predominant virus strain in August 2022. The epidemic curves of COVID-19 in the first to the fourth wave as well as from the fifth wave onwards are illustrated in Figure 2 and Figure 3, respectively.

## 4. Evolving SARS-CoV-2 VOCs in Hong Kong

Diverse SARS-CoV-2 genomes were identified among imported infections during the first to the fourth wave [51]. The importation of different lineages, variants, and subvariants resulted in transmission chains in the local community even before declaration of the COVID-19 pandemic [52]. Except for the first and second waves, local infections were dominated by a single lineage during each wave. Based on whole-genome sequencing analysis of virus strains, 96.6% (259/268) in the third wave and 100% (73/73) in the fourth wave were B.1.1.63 and B.1.36.27 lineages, respectively. Although B.1.1.63 lineage was first detected in an imported case 2 weeks before the beginning of the third wave, B.1.36.27 lineage had circulated in Hong Kong for 2 months prior to the fourth wave [53]. During the window phase in which there was no evidence of local transmission of COVID-19 (near-zero situation), 212 virus strains from imported cases were subjected to next-generation sequencing, which revealed 42 Kappa and 70 Delta variant cases [54]. By February 2022, the Omicron variant had become the predominant virus strain, causing explosive community transmission and outbreaks in high-rise residential buildings during the fifth wave [50,55]. In addition, probable animal-to-human transmission of SARS-CoV-2 Delta variant AY.127 was documented to cause a pet shop-related COVID-19 outbreak in Hong Kong during the early phase of the fifth wave [56]. This hamster-related outbreak led to a cluster of at least 88 cases in our epidemiological investigation [57]. The evolution of the epidemic due to SARS-CoV-2 VOCs in Hong Kong is illustrated in Figure 4.

## 5. Infection Control and Preparedness for COVID-19 in Hong Kong

### 5.1. Background

In Hong Kong, the SARS-CoV-1 outbreak took place in 2003. A total of 1755 persons were infected and 299 (17.0%) persons died [58]. Among the infected, 386 (22.0%) were healthcare workers, with eight fatalities, including four doctors, one nurse, and three healthcare assistants. In addition to reviewing the lessons learned from the scientific perspective [15], an independent commission of inquiry, namely, the “Select Committee to inquire into the handling of the Severe Acute Respiratory Syndrome outbreak by the Government and the Hospital Authority,” was set up by the Legislative Council of the Hong Kong Special Administrative Region, China, in order to investigate the performance and accountability of the government and the Hospital Authority (the governing body of all 43 public hospitals responsible for 90% of inpatient service in Hong Kong) and their officers at the policy-making and management levels [59]. The Chief Executive of the Hong Kong Special Administrative Region also announced on 28 May 2003 the setting up of a SARS Expert Committee to (i) review the work of the government, including the Hospital Authority, in the management and the control of the outbreak; (ii) examine and review the capabilities and structure of the healthcare system in Hong Kong and the organization and operation of the Department of Health and the Hospital Authority in the prevention and management of infectious diseases such as SARS; and (iii) identify lessons to be learned and make recommendations on areas of improvements in order to better prepare our system for any future outbreaks [60,61]. Members of the committee were selected for their wide range of experience in their respective fields, which included health systems, public health, epidemiology and communicable disease control, medical expertise, and hospital management and operation.

There were 44 main recommendations by the SARS Expert Committee, as a result of which the Centre for Health Protection was established in June 2004 to protect the community from emerging or evolving public health threats [62]. Preparedness plans for emerging infectious diseases have been formulated over the years. When the outbreak of SARS-CoV-2 occurred in Hong Kong, various control measures could be implemented in a timely manner, including epidemiological investigation for the potential source of acquisition by contact tracing, regardless of symptomatic or asymptomatic cases. Risk communication to the public during the COVID-19 pandemic included daily press conferences on the latest situation and public health control measures and uploading the data and education materials to the website.

A hospital-based approach was adopted to isolate all suspected or confirmed COVID-19 patients in healthcare facilities, including airborne infection isolation rooms (AIIRs) in hospitals or community isolation and treatment facilities, during the first to the fourth wave of COVID-19 in Hong Kong. This approach of institutional containment likely minimized community transmission of COVID-19. The strategy was effective in protecting the healthcare system from being overwhelmed by uncontrolled transmission of the virus in the community. With 5% of COVID-19 patients eventually becoming critically ill, the healthcare system might have become paralyzed if the burden of infection in the community were high [63]. Prevention of nosocomial transmission of COVID-19 and protection of healthcare workers from SARS-CoV-2 remained the top priority, which dictated regular review and escalation of infection control measures during the evolution of the COVID-19 pandemic.

### 5.2. Infection Control for COVID-19 in the Hospitals

#### 5.2.1. Active Surveillance

Active surveillance for patients upon admission was progressively stepped up from risk-based screening to universal admission screening, aiming at early identification and isolation of COVID-19 patients in AIIRs (Table 4). Patients fulfilling clinical and epidemiological criteria were isolated in the AIIRs and screened for SARS-CoV-2 by reverse-transcription–real-time polymerase chain reaction (RT-PCR) using nasopharyngeal specimens or saliva [64,65].

Saliva was adopted as one of the diagnostic specimens for the detection of SARS-CoV-2 RNA in Hong Kong during the early phase of COVID-19. SARS-CoV-2 RNA was detected in self-collected saliva specimens in 91.7% (11/12) of our early COVID-19 cases [66]. Using saliva specimens, the temporal profile of viral load during the clinical course could also be demonstrated [67]. The diagnostic sensitivity of saliva is comparable to nasopharyngeal specimens using an automated point-of-care molecular assay in a local evaluation [68], as subsequently confirmed in a meta-analysis [69]. With diurnal variation in viral shedding from the upper respiratory tract, saliva was recommended to be taken preferably in the early morning to maximize the yield for screening purposes in the community [70]. Saliva was considered a promising noninvasive specimen for diagnosis and monitoring of COVID-19 patients. More importantly, it reduced the risk of exposure among healthcare workers in collecting nasopharyngeal and other respiratory specimens.

#### 5.2.2. Training of Healthcare Workers

Provision of just-in-time education of infection control training was organized by the hospital infection control team. Staff forum and department visits were organized to update infection control knowledge, enforce hand hygiene practice using alcohol-based hand rub, and promote the use of surgical masks for both patients and healthcare workers. Furthermore, every healthcare worker who was rotated to care for COVID-19 patients in AIIRs was required to receive mandatory training by infection control nurses. The training program included the donning and doffing of personal protective equipment (PPE), the collection of respiratory specimens, proper handling and packing of clinical specimens inside the AIIR, and performing aerosol-generating procedures in accordance with the evolving infection control guidelines [71,72]. After receiving the training, each healthcare worker was required to demonstrate competence on donning and doffing of PPE, including surgical respirator, cap, face shield, gown, and gloves. The concept of directly observed donning and doffing of PPE was introduced, where the supervisor in the ward would ensure compliance with proper PPE procedures upon entering and leaving the AIIR. Infection control nurses also performed regular and ad hoc audits to enforce compliance with infection control practices. Hand hygiene and mask compliance among healthcare workers were found to be highly satisfactory during the COVID-19 pandemic [73,74].

#### 5.2.3. Environmental Control

Environmental control was another important aspect to reduce nosocomial transmission of COVID-19. Training of cleaning staff, especially those responsible for environmental disinfection of AIIRs, with respect to proper procedures and the use of appropriate disinfectants, was conducted by infection control nurses. A total of 377 environmental samples from AIIRs housing 21 RT-PCR-confirmed COVID-19 patients were collected during 76 episodes of room visits before daily environmental disinfection. Only 19 (5.0%) of 377 samples were RT-PCR positive from the environment [75], which was comparatively lower than isolation rooms in China (25%) and South Korea (14% to 18%) [76,77]. The use of surgical masks by COVID-19 patients, even though they were isolated in the AIIRs, might have reduced the extent of environmental contamination. Patient transfer between wards could potentially have increased the risk COVID-19 transmission within the hospital [78]. However, intra-hospital transfer of COVID-19 patients was unavoidable if the case required certain procedures, including radiological examination such as computerized tomography scanning. In addition to routine preparations made by the radiology department, the radiological examination was preferably arranged after office hours. Senior infection control nurses would provide on-site support and coordination for the entire process of patient transfer, including provision of a designated path, supervision of the donning and doffing of PPE among healthcare workers and security personnel, and monitoring the process of environmental disinfection along the path of patient transfer and in the examination room of the radiology department.

#### 5.2.4. Air Sampling for SARS-CoV-2 RNA

To assess the risk of COVID-19 transmission in the hospital, air sampling for SARS-CoV-2 RNA was conducted in AIIRs with 12 air changes per hour during the evolution of the COVID-19 pandemic. The first imported COVID-19 case admitted into an AIIR had a moderate level of viral load of 3.3 × 10^6^ copies/mL in pooled nasopharyngeal and throat swabs and 5.9 × 10^6^ copies/mL in saliva. Air samples for SARS-CoV-2 RNA were collected from the AIIR using air sampler SAS Super ISO 180 model 86834 (VWR International PBI Srl, Milan, Italy). The air sampler was perpendicularly positioned at a distance of 10 cm at the level of the patient’s chin, and 1000 L of air at a rate of 180 L per minute were collected for each plate containing 3 mL of viral transport medium (VTM). The patient was instructed to perform four different maneuvers (i.e., normal breathing, deep breathing, speaking “1, 2, 3” continuously, and coughing continuously) while wearing or not wearing a surgical mask with the ASTM F2100 level 1 standard. The VTM was transferred to the laboratory within 2 h for RT-PCR test. All eight air samples were negative for SARS-CoV-2 RNA [64].

Subsequently, the protocol of air sampling was modified for another six patients singly isolated in an AIIR during the first and second waves of COVID-19 by using an umbrella fitted with a transparent plastic curtain as an air shelter to cover the patient during sample collection in order to increase the proportion of exhaled air sampled and to reduce the amount of dilution by environmental air from the air ventilation system with 12 air changes per hour. Air samples of patients inside this air shelter were collected using the Sartorius MD8 air-scan sampling device (Sartorius AG, Göttingen, Germany) with sterile gelatin filters 80 mm in diameter and a pore size of 3 μm (type 17528-80-ACD) (Sartorius AG). All air samples were negative for SARS-CoV-2 RNA using the same maneuvers as described above [75]. The negative findings of air samples in the initial phase of the COVID-19 pandemic reassured our healthcare workers that the risk of airborne transmission of SARS-CoV-2 from patient to staff was negligible in AIIRs where healthcare workers donned appropriate PPE. Our finding contrasts with previous air sampling results for patients infected with the ancestral strain [79,80,81]. Most negative studies had <3000 L of air collection [64,75,79,80,81], whereas a few studies with detectable viral RNA had >5000 L of air collection [82,83]. The results of air sampling appear to be determined by the volume of air collection, which might be limited by the intrinsic characteristic of the air samplers.

Therefore, AerosolSense Sampler (Thermo Fisher Scientific Inc., Waltham, MA, USA) was applied because a larger volume of air could be collected at a flow rate of 200 L per min. In fact, air dispersal of other respiratory viruses such as parainfluenza virus, respiratory syncytial virus, adenovirus, and rhinovirus have been demonstrated in the AIIR using AerosolSense Sampler [84]. We further investigated an asymptomatic COVID-19 patient infected with a PANGO lineage B.1.525 virus, singly isolated in an AIIR, before the onset of the fifth wave. The patient had a moderate level of viral load (6.8 × 10^6^ copies/mL in a nasopharyngeal swab sample). SARS-CoV-2 RNA was detected by air sampling (96,000 L air collected over 8 h) at a concentration of 0.005 and 0.009 copies/L, respectively, while the patient was and was not wearing a surgical mask [85]. Given the low level of air dispersal of SARS-CoV-2 RNA, the risk of inhalation of SARS-CoV-2 by healthcare workers in hospital AIIRs was considered extremely low.

During the fifth wave, general wards were temporarily converted into negative-pressure wards (NPWs) by installing mobile modular high-efficiency particulate arrestance filter units and exhaust fans in each cubicle to increase air changes per hour from 6 to 10 to accommodate COVID-19 patients. We collected air samples in the center of these NPWs for 1 to 6 h, with a total volume of 12,000 L to 72,000 L of air. Air dispersal of SARS-CoV-2 RNA was evidenced by almost 80% of air samples being positive while the patients in these NPWs were infected with the Omicron sub-lineage BA.2.2 [86]. Our serial monitoring of air sampling during the evolution of the COVID-19 pandemic demonstrates that SARS-CoV-2 variants had progressively increasing capability for airborne transmission. In addition, air dispersal of other respiratory viruses and multiple-drug-resistant bacteria were also evaluated during the COVID-19 pandemic [84,87,88]. The findings of air dispersal of SARS-CoV-2 as well as other pathogens facilitated risk assessment to guide implementation of infection prevention and control measures.

### 5.3. Relieving the Burden of COVID-19 Patients in the Hospitals

#### 5.3.1. Proactive Screening of High-Risk Groups in Quarantine Centers

The hospital-based approach to admit all suspected and confirmed COVID-19 patients into AIIRs in hospitals posed great pressure on the bed occupancy, especially during the early phase of the COVID-19 outbreak when Hong Kong residents returned from overseas. Quarantine centers were set up by the government for quarantine of close contacts of confirmed cases and high-risk returnees, including one set up in a newly completed public housing estate, Chun Yeung Estate (CYE), which could provide 3121 residential units for the purpose of quarantine. Returning travelers from high-risk locations such as the *Diamond Princess* cruise ship and Hubei province, China, were accommodated in CYE. On 21 February 2020 (day 53), 215 SARS-CoV-2 RT-PCR-negative passengers of the *Diamond Princess* cruise ship, returning to Hong Kong from Yokohama, Japan, on two chartered flights arranged by the Hong Kong Government, were quarantined for 14 days in CYE. The quarantined persons underwent serial SARS-CoV-2 RT-PCR and serology testing on day 0 (baseline), day 4, day 8, and day 12, and were discharged from the quarantine center on day 14 if all the test results were negative for SARS-CoV-2. The clinical and virological characteristics of this cohort have been reported previously [89]. Since 4% (4/215) of quarantined persons were eventually positive for SARS-CoV-2 by RT-PCR, it was decided that 469 Hong Kong residents evacuated from Hubei province, China, on four different chartered flights on 4 and 5 March 2020 would be admitted to the quarantine center in CYE for 14 days. Seventeen (4%) of 452 persons who consented to blood testing were seropositive with the enzyme immunoassay or microneutralization test [90]. Our healthcare workers provided outreach service to screen for COVID-19 patients among these returning travelers. Only persons who had positive RT-PCR results during the quarantine period were referred to the hospital for further isolation and management.

#### 5.3.2. Setting Up Temporary Test Centers for Suspected COVID-19 Patients

Symptomatic inbound travelers fulfilling the reporting criteria (Table 4) would be referred to hospitals for isolation and testing. The innovative idea of setting up a temporary test center at the exhibition hall of the AsiaWorld-Expo (AWE) within the Hong Kong International Airport complex aimed at detecting COVID-19 infection among returning travelers aged 12 to 60 years who were clinically stable but had respiratory symptoms upon arrival. The logistics and workflow were based on the infection control principle that the hall was divided into two zones, a clean zone and a patient zone, with clear demarcation by double-fencing. The clean zone included a staff area, a gowning area, and a central command center. The patient zone included areas for patient registration, assessment, collection of clinical specimens, and waiting for diagnostic test results. Unidirectional workflow was assigned for both patients and healthcare workers. Patients who tested positive for SARS-CoV-2 were referred to the hospital for further management. The temporary test center for symptomatic travelers was under the governance of the Hospital Authority, operated by healthcare workers given just-in-time infection refresher training on site. This test center commenced service within 48 h of preparation on 20 March 2020 (day 81) [91]. During the second wave, from 20 March 2020 to 19 April 2020 (day 111), 1210 symptomatic cases that met the criteria for hospitalization under the hospital-based approach were tested at the temporary test center. Eighty-eight (7.3%) of these 1210 patients tested positive for SARS-CoV-2 [92].

#### 5.3.3. Setting Up Community Isolation and Treatment Facilities for COVID-19 Patients

With the further surge in cases during the third wave of COVID-19 as a result of increasing community outbreaks, community isolation and treatment facilities were set up to divert clinically stable patients from hospitals. The community isolation facility was opened at Lei Yue Mun Park and Holiday Village on 24 July 2020 (day 207) [93]. Bungalows were temporarily built on two basketball courts to accommodate 120 and 250 patients in two wings at sites A and B, respectively. Patients could be transferred from hospitals to the bungalow if they were aged <50 years, were independent for activities of daily living, had no major comorbidity, were afebrile for 48 h, had no diarrhea, had normal or improving trends of hematology and biochemistry test profiles, and were not on antiviral or oxygen therapy. As for the community treatment facility, it was established at AWE and commenced operation on 1 August 2020 (day 215). Two halls at the AWE, hall 1 (500 beds; floor area 10,880 m^2^ and ceiling 19 m) and hall 2 (400 beds; floor area 10,100 m^2^ and ceiling 10 m), were converted into open-cubicle bays. Newly diagnosed COVID-19 patients aged 18 to 60 years were triaged for admission to the community treatment facility after medical and nursing assessment, blood tests, and chest radiography. Medically fit patients with clear chest radiography and oxygen saturation ≥ 96% of room air were managed in the community treatment facility until discharge [93]. This facility was built in a convention center, similar to the setup of temporary or shelter hospitals in Wuhan, China [94,95]; Singapore [96]; the United Kingdom [97,98]; and the United States of America [99]. Although the community isolation facility at Lei Yue Mun Park and Holiday Village was closed on 17 August 2020 (day 231), the community treatment facility at AWE relieved the occupancy of AIIRs by diverting COVID-19 patients from the hospitals during the third wave (1 August 2020 to 18 September 2020, serving for 49 days) and the fourth wave (25 November 2020 to 12 March 2021, serving for 108 days).

During the fifth wave (primary Omicron wave) of COVID-19 due to Omicron BA.2, there were more than 10,000 new cases per day in late February 2022. In addition to the AWE (2 January 2022 to 4 May 2022, serving 123 days), public rental buildings (~3000 residential flats), Fangcang shelters (~20,000 beds), and the Kai-Tak cruise terminal (~1000 beds) were recruited as community treatment facilities, in view of the explosive COVID-19 outbreaks in the community [50,55,100]. In anticipation of ongoing transmission of COVID-19, community isolation facilities were built in Penny’s Bay (around 7000 units and 14,000 beds) [101] and Kai Tak (around 2700 units) [102], whereas the community treatment facility at AWE was re-opened for the secondary Omicron wave (22 August 2022 to 29 September 2022, serving 39 days).

### 5.4. COVID-19 Infection among Healthcare Workers

Infection and death among healthcare workers due to COVID-19 was reported in the early phase of the pandemic [103]. As of 8 May 2020 (day 130), a total of 152,888 infections and 14,113 deaths were reported among healthcare workers globally. Healthcare workers constituted 0.52% of COVID-19 deaths in a systematic review [104]. The WHO further estimated that between 80,000 and 180,000 healthcare workers could have died from COVID-19 in the period between January 2020 and May 2021, converging in a scenario of 115,500 deaths [105]. Although healthcare workers may acquire SARS-CoV-2 in both hospitals and the community [106], Hong Kong aimed at minimizing nosocomial infection of COVID-19 among healthcare workers [107]. Zero nosocomial COVID-19 infection among healthcare workers in the public system was achieved in the first 300 days, with a total of 78,834 COVID-19 patient days, through the implementation of a multipronged infection control strategy, including active surveillance, universal masking for patients and healthcare workers, provision of just-in-time infection control training, practicing directly observed donning and doffing of PPE, and diverting COVID-19 patients from hospitals to quarantine camps, community isolation and treatment facilities, as described above [65]. By day 300 (25 October 2020), a total of 38 healthcare workers had acquired SARS-CoV-2 in the community. The proportion of healthcare worker infection in our public healthcare system was significantly lower than that of the general population (0.04%, 38/88,960 vs. 0.07%, 5296/7403,100; *p* < 0.001), suggesting high infection control alertness among our healthcare workers during the first to third wave of COVID-19 in Hong Kong. The first nosocomial outbreak of COVID-19 was reported on 22 December 2020 (day 358) in a regional hospital, with a superspreading event due to possible airborne transmission involving 12 patients and nine healthcare workers. Whole-genome sequencing revealed that the nosocomial outbreak was attributed to SARS-CoV-2 lineage B.1.36.27 (GISAID clade GH), which was predominant in the fourth wave of COVID-19 in Hong Kong [108].

During the fifth wave, it was reported that 12,554 healthcare workers had been infected with COVID-19, overwhelming the public healthcare system as of 11 March 2022 (day 802) [109]. The proportion of healthcare worker infection was paradoxically higher than that of the general population during the corresponding period (14.1%, 12,554/88,960 vs. 8.7%, 646,800/7403,100; *p* < 0.001). The discrepancy may be related to the mandatory daily COVID-19 rapid antigen testing among healthcare workers, such that asymptomatic cases were also detected [110]. Frontline healthcare workers were more likely to report a positive COVID-19 test compared with community individuals in the United Kingdom and the United States [111]. For our infected healthcare workers, it was difficult to determine whether they had been infected in the hospital or in the community, because our staff was not working under closed-loop management like in mainland China [112]. There were no reported deaths among our healthcare workers during the COVID-19 pandemic.

## 6. Public Health Measures for COVID-19 in Hong Kong

### 6.1. Background

To quantify the public health measures, a stringency index, namely, the Oxford COVID-19 Government Response Tracker, was established for COVID-19 [113]. This index is a composite measure based on the policy responses that governments have implemented to tackle COVID-19. The data captured government policies related to closure and containment, and health and economic policy for more than 180 countries from 1 January 2020 (day 2) onwards and are accessible in the public domain [114]. The detailed methodology has been published [115]. At the time of declaration of the COVID-19 pandemic by the WHO on 11 March 2020 (day 72), the incidence of infection per 10,000 population and the number of COVID-19 deaths in Hong Kong were comparatively lower than those of the other countries in the Western Pacific, European, and American regions [107]. Hong Kong appeared to have fared better in controlling COVID-19 with public health measures [116]. The COVID-19 stringency index of Hong Kong and other countries during the evolution of the pandemic is illustrated in Figure 5.

### 6.2. Universal Masking and Social Distancing for COVID-19

Community-wide masking was practiced by the Hong Kong population at an early stage of the epidemic. The compliance of face mask usage was 96.6% (range: 95.7% to 97.2%) in all 18 administrative districts in Hong Kong [117]. This may be one of the reasons why the incidence of COVID-19 in Hong Kong (129.0 per million population) was significantly lower than that of Spain (2983.2), Italy (2250.8), Germany (1241.5), France (1151.6), the US (1102.8), the UK (831.5), Singapore (259.8), and South Korea (200.5) within the first 100 days [117]. Mandatory wearing of masks in public areas was further enacted by the law “Cap. 599I Prevention and Control of Disease (Wearing of Mask) Regulation,” which went into effect on 15 July 2020 [118]. Without a total lockdown of the city or complete border closure, the aim was the implementation of precise social distancing to reduce community transmission of COVID-19. School was suspended and gradually converted to virtual class during most of the time during the first to the fourth wave of COVID-19. Work-from-home for non-essential services of civil servants was intermittently implemented. Temporary closure of community facilities such as libraries, sports centers, and leisure facilities such as cinemas, karaokes, and bars, as well as restrictions on operational hours of restaurants, was implemented during the surge of COVID-19 cases in the community [119]. Multivariate analysis of computational simulation results using the Morris Elementary Effects Method suggested that if a sufficient proportion of the population used surgical masks and followed social distancing regulations, COVID-19 could be controlled without requiring a lockdown [120].

### 6.3. Quarantine Measures for Inbound Travelers

A key component of an elimination strategy for COVID-19 was the prevention of the importation of COVID-19 cases. The quarantine measures for inbound travelers evolved to accommodate emerging SARS-CoV-2 VOCs. From the beginning, persons returning from mainland China had to undergo compulsory home quarantine for 14 days after 8 February 2020 (day 40) [121]. With the evolving pandemic, inbound travelers arriving from the specific high-risk overseas areas in the past 14 days were required to stay in a quarantine center from 1 March 2020 (day 62) [122,123] or subjected to compulsory home quarantine on 14 March 2020 (day 75) [124]. Subsequently, inbound travelers arriving from more high-risk areas were required to be quarantined for 14 days in quarantine centers after 11 May 2020 (day 133) or hotels after 25 July 2020 (day 208) [125]. On 13 November 2020 (day 319), all travelers to Hong Kong were mandated to quarantine in designated quarantine hotels for 14 days [126], and the quarantine period was further extended to 21 days on 25 December 2020 (day 361) [127]. The duration of hotel quarantine was shortened from 21 days to 14 days on 5 February 2022 (day 768) [128] and further shortened to 7 days on 1 April 2022 (day 823) for travelers vaccinated with two doses [129]. On 12 August 2022 (day 956), quarantine for inbound persons was relaxed with the implementation of the “3 + 4” model, i.e., compulsory quarantine in designated hotels for three days, followed by medical surveillance for four days, with multiple tests during the period and the monitoring period thereafter [130]. With effect from 26 September 2022 (day 1001), inbound persons were no longer required to undergo compulsory quarantine at designated quarantine hotels under the “0 + 3” model, i.e., three days of medical surveillance, followed by a four-day self-monitoring period [131]. The number of passenger arrivals at Hong Kong International Airport during the evolving quarantine policy is shown in Figure 6.

The designated quarantine hotels were not equipped with the standard of air ventilation systems adopted in the AIIRs of hospitals. With the increasing potential of airborne transmission of SARS-CoV-2 VOCs, recurrent episodes of cross-infection among residents in designated quarantine hotels were reported [132,133], leading to the spread of the Beta variant in the community [134]. In fact, the explosive outbreak in the fifth wave of COVID-19 in the community could also be traced to cross-infection in a designated quarantine hotel [55].

### 6.4. Extensive COVID-19 Testing in the Community

In addition to the progressive enhancement of COVID-19 testing in the hospitals from a risk-based approach to universal admission screening, widespread utilization of the RT-PCR test for SARS-CoV-2 was implemented in stages at the airport for inbound travelers, in government outpatient clinics for symptomatic cases, and in community testing centers for the general population with risk of exposure to COVID-19 [134]. Early recognition of asymptomatic or mildly symptomatic patients for isolation might have reduced the risk of community transmission. Compulsory testing for COVID-19 in the community was enacted by law, The Prevention and Control of Disease (Compulsory Testing for Certain Persons) Regulation, Cap. 599J, in Hong Kong [135]. Any persons who had visited public or private premises and resided in buildings where there was outbreak or transmission of COVID-19 as evidenced by epidemiological investigations or sewage surveillance were informed of the need to undergo COVID-19 testing by a compulsory testing notice. Compulsory testing operations in specified restricted geographic areas were also implemented for the control of COVID-19, as provided by restriction-testing declaration under Hong Kong legislation [136]. With the implementation of compulsory testing notices and restriction-testing declarations, mass testing of 0.81 million members of the population, along with phylogeographic analysis of positive cases, successfully controlled the community outbreak of the Beta variant in Hong Kong [134]. The number of COVID-19 tests performed in different categories during the COVID-19 pandemic is illustrated in Figure 7.

Restriction-testing declaration for COVID-19 testing, particularly targeting residential buildings, was implemented during the fifth wave (primary and secondary Omicron waves) of COVID-19. From 31 December 2021 (day 732) to 25 September 2022 (day 1000), a total of 606,822 residents in 473 residential buildings were tested, of whom 27,726 (4.6%) tested positive for SARS-CoV-2 (Figure 8). Confirmed COVID-19 cases were required to isolate at community isolation and treatment facilities or in hospitals. The positive detection rate of SARS-CoV-2 in each high-rise residential building was as high as 30% during the peak of the fifth wave [50]. Vertical airborne transmission of SARS-CoV-2 was suspected among residents, as evidenced by the clustering of cases along vertically aligned flats with a shared drainage stack and lightwell [50].

However, no air sampling was performed to demonstrate the presence of SARS-CoV-2 RNA in high-rise residential buildings to confirm the postulation of airborne transmission. On the other hand, air dispersal of SARS-CoV-2 RNA was confirmed in the hospital setting during the handling of recurrent blockages of sewage pipes in an isolation facility designated for COVID-19 patients. During replacement of sewage pipes, infection control nurses supervised the work process and collected air samples on cutting the sewage pipes, which was considered a mechanical aerosol-generating procedure. The air sampler was placed 50 cm away from the pipe cutter. One thousand liters of air (rate 40 L/min) were collected by a gelatin filter for the first 25 min of work (air sample 1), and another 1000 L of air were collected by another gelatin filter for the next 25 min of work (air sample 2). Air dispersal of SARS-CoV-2 RNA was detected in both air sample 1 (17.5 copies/mL) and air sample 2 (16.5 copies/mL) [137].

### 6.5. COVID-19 Vaccination in the Community

The COVID-19 vaccination program was officially launched on 26 February 2021 (day 424) by the Government of the Hong Kong Special Administrative Region. Two formulations of COVID-19 vaccines were available: CoronaVac, an inactivated whole-cell vaccine, Sinovac Biotech (Hong Kong) Limited, and Comirnaty, a BNT162b2 mRNA vaccine, BioNTech, Fosun Pharma, in collaboration with the German drug manufacturer. Vaccination was provided for free to all eligible persons, who could be vaccinated in community vaccination centers, private clinics of medical practitioners, and public outpatient clinics and hospitals. To increase the uptake of COVID-19 vaccination, promotion to healthcare workers as well as the general population was implemented. The strategy of personal coaching was undertaken to enhance vaccination update among healthcare workers, similar to the strategy for the promotion of influenza vaccination [138]. An official website was set up to deliver updated information related to COVID-19 vaccination in multiple languages [139]. The daily number of COVID-19 vaccine doses administered was made available in the public domain of “Our World in Data” [140], and subsequently analyzed by the Centre for Health Protection for daily reporting from 5 March 2022 (day 796) [141].

Vaccine hesitancy is a major challenge in promoting the COVID-19 vaccination campaign in the locality. In a cross-sectional survey of a random sample of 1501 Hong Kong residents aged 18 years or older in April 2020, only 45.3% of the participants intended to vaccinate against SARS-CoV-2 when available [142]. Subsequently, an online survey was conducted during an early stage of a community-based COVID-19 vaccination campaign in Hong Kong. Among the 883 participants (67.5% females, 54.5% aged 18–39), 30.6% had low vaccine hesitancy, 27.4% had high vaccine hesitancy, and 27.5% had vaccine rejection [143]. Vaccine hesitancy is not an uncommon phenomenon in the general population of developed and developing countries [144,145,146]. Vaccine hesitancy was also observed in healthcare workers as well as medical students [147,148,149]. The most predominant predictors of vaccine hesitancy were a lower perceived risk of getting infected, a lower level of institutional trust, not being vaccinated against influenza, lower levels of perceived severity of COVID-19, or stronger beliefs that the vaccination would cause side effects or be unsafe, as illustrated in a systemic review on the global predictors of COVID-19 vaccine hesitancy [150]. Institutional trust is an important factor that may increase vaccine willingness and uptake. Trust in the government and civil societies tended to strengthen the positive effect of information overload and reduce the negative impact of misinformation on vaccine willingness and uptake [151]. Vaccine complacency, an unintended side effect of successful control of COVID-19, is another important factor that adversely affects the intention to receive the vaccine in Hong Kong [152]. In a cross-sectional online survey of 1205 nurses, less than two-thirds of nurses intended to take the COVID-19 vaccine when available. Stronger COVID-19 vaccination intention was associated with less complacency, together with younger age, more confidence, and more collective responsibility [153]. Similarly, lower complacency, greater anxiety, confidence in the vaccine, and collective responsibility contributed to a greater likelihood of intended vaccination in a population-based online survey in Hong Kong [154]. Other regions using strategies to suppress and almost eliminate COVID-19 including Taiwan, Macau, the Chinese mainland, Tonga, and Western Australia have also experienced vaccine complacency [155].

Therefore, an administrative measure, the Vaccine Pass, was implemented to overcome vaccine hesitancy in Hong Kong. The Vaccine Pass was incorporated in a mobile app known as LeaveHomeSafe, which could be downloaded from the App Store, Google Play, AppGallery, and APK File [156]. To protect the unvaccinated and to boost vaccination coverage in the community, the Vaccine Pass arrangement was implemented for entry into all catering business and scheduled premises from 24 February 2022 (day 422) [157,158]. The Vaccine Pass was introduced for local residents in stages (Table 5). The daily number of COVID-19 vaccines administrated in relation to the implementation of the Vaccine Pass is illustrated in Figure 9.

Up to 25 September 2022 (day 1000), 6,683,654 (90.3%) and 5,351,219 (72.3%) of the general population had received the second and third doses of the COVID-19 vaccine, respectively, in Hong Kong. Regarding the population with two doses of vaccination, the age-specific vaccination rate remained low at the extremes of age (Figure 10), which led to over 9000 deaths during the primary Omicron wave. As for vaccine efficacy, in a study covering the period between 31 December 2020 and 16 March 2022, during which 13.2 million vaccine doses had been administered, two doses of either vaccine formulation were shown to protect against severe disease and death within 28 days after confirmation of infection by a positive test. Higher effectiveness was demonstrated among the subgroup of adults aged 60 years or older who had received BNT162b2 (vaccine effectiveness 89.3%) when compared with CoronaVac (69.9%). Three doses of either vaccine offered very high levels of protection against severe or fatal outcomes (97.9%) [159]. COVID-19 vaccines have been shown to be safe in patients with cancer and chronic diseases and in people aged 60 years or older in Hong Kong [160,161,162,163].

### 6.6. Other Non-Pharmaceutical Intervention in the Community

As mask-off activities such as dining and drinking in restaurants are associated with COVID-19 transmission [117], legislation was implemented on air-change requirements or utilization of air purifiers in dine-in catering premises [164]. Specifically, catering business operators were required to ensure six or more air changes per hour in the seating area. Otherwise, air purifiers were to be installed as an alternative measure to reduce the risk of airborne transmission of SARS-CoV-2 [165].

## 7. The Way Forward

Although the stringent control measures of universal masking, social distancing, testing for isolation and quarantine, contact tracing, and border testing with quarantine worked well for 2 years with previous SARS-CoV-2 variants or lineages, such measures failed to control the Omicron variant in Hong Kong. The global spread of the more benign Omicron variant might be the beginning of the end for the COVID-19 pandemic [166,167], given that a high proportion of the global population has some immunity from natural infection, vaccination, or hybrid immunity. The risk of COVID-19 related death associated with Omicron was comparatively lower than that of the Delta variant [168,169] and was also lower than the previous variants [170], even during the pre-vaccination era [171]. Paradoxically, the crude population mortality rate for COVID-19 in Hong Kong (37.7 per million) during the peak level of the fifth wave, caused by the BA.2 sub-lineage, was among the highest reported worldwide because of the low COVID-19 vaccination coverage in the elderly population [172]. The severe community outbreak of Omicron BA.2 could be attributed to the far higher airborne transmissibility as well as the lower population immunity [173]. The overall seropositive rate of IgG against the receptor-binding domain of SARS-CoV-2 increased from 52% in December 2021 to 89% in May 2022, at the end of the fifth wave of COVID-19 in Hong Kong [174]. Hybrid immunity has been established locally and the case fatality rate of COVID-19 during the secondary Omicron wave, predominantly caused by Omicron BA.5, was about 0.1%, which was comparable to the case fatality rate of seasonal influenza. This could be attributable to the marked increase in vaccination rate and early antiviral treatment. However, the case fatality rate among unvaccinated persons still remained higher at 0.6% overall and 9% among elderly aged 80 years or above. After analyzing scientific data and striking a balance among factors such as transmission risks, the public health policy of compulsory quarantine in designated quarantine hotels for all inbound travelers was no longer required from 26 September 2022 (day 1001), facilitating international travel and recovery of the economy while being able to contain the spread of infection. In Hong Kong, the healthcare services have gradually resumed normalcy. Healthcare workers were fully vaccinated and are well trained for COVID-19. The modalities of pharmaceutical intervention, including remdesivir, nirmatrelvir/ritonavir, molnupiravir, baricitinib, tocilizumab, and tixagevimab co-packaged with cilgavimab were made available to patients according to local studies [175,176,177,178,179,180] and in accordance with the evolving treatment guidelines [181,182]. The government will continue to adjust the anti-pandemic strategy based on scientific data and evidence [183].

Analogous to the influenza virus with the phenomenon of antigenic drift and shift [184], new variants of SARS-CoV-2 are expected to develop with time. Since September 2022, Omicron sub-lineages BA.2.75, BF.7, and BA.4.6 have been reported to be increasing in Asia (India, Nepal, Singapore), Europe (Belgium, France, German), and the United States, respectively [185]. These three sub-lineages could substantially escape neutralizing antibodies induced by vaccination or previous infection, or both [186]. International collaboration is required to closely monitor the evolving SARS-CoV-2 variants and see whether the T lymphocyte response induced by infection, vaccination, or hybrid immunity will continue to turn this virulent pandemic virus into an endemic benign respiratory virus. Infection control and public health measures remain the most important non-pharmaceutical interventions against COVID-19 and any other emerging infectious diseases.

## Figures and Tables

**Figure 1 viruses-14-02519-f001:**
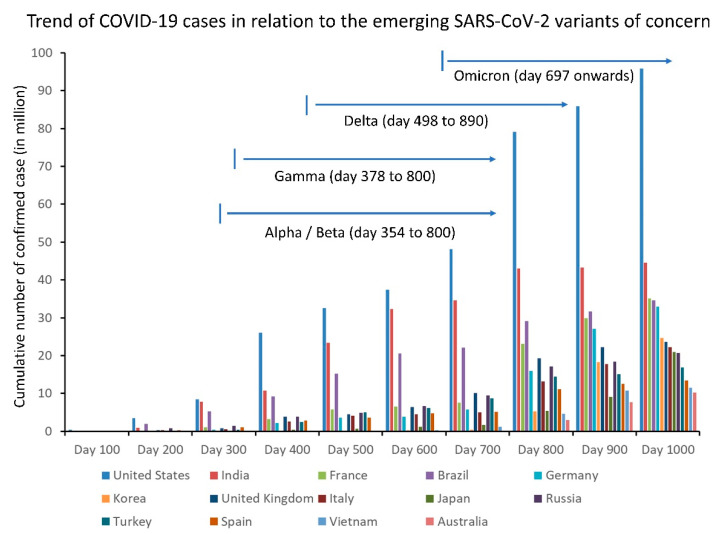
Cumulative number of COVID-19 cases in 14 countries reporting more than 10 million cases as of 25 September 2022 (day 1000).

**Figure 2 viruses-14-02519-f002:**
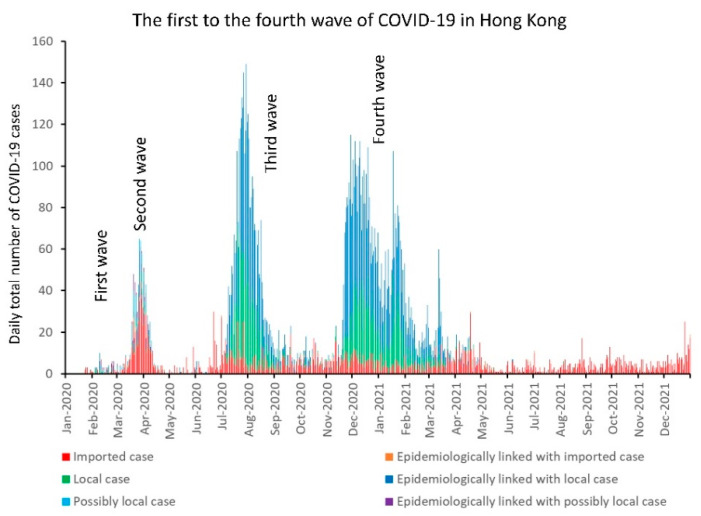
Epidemic curve of the first to the fourth wave of COVID-19 in Hong Kong. An imported case is defined as a case having traveled to a country or area with community transmission of COVID-19 during the incubation period (IP) (i.e., operationally defined as 7 days before symptom onset or a positive test for an asymptomatic case) or tested positive during compulsory quarantine without having any activity in the local community. A local case is defined as a case without travel history during the IP. A possibly local case is defined as those having traveled to a country or area not known to have community transmission within the IP (this category was historically used in the first two waves in early 2020 only). Imported cases, local cases, and cases epidemiologically linked with local cases constituted the main burden of COVID-19 cases.

**Figure 3 viruses-14-02519-f003:**
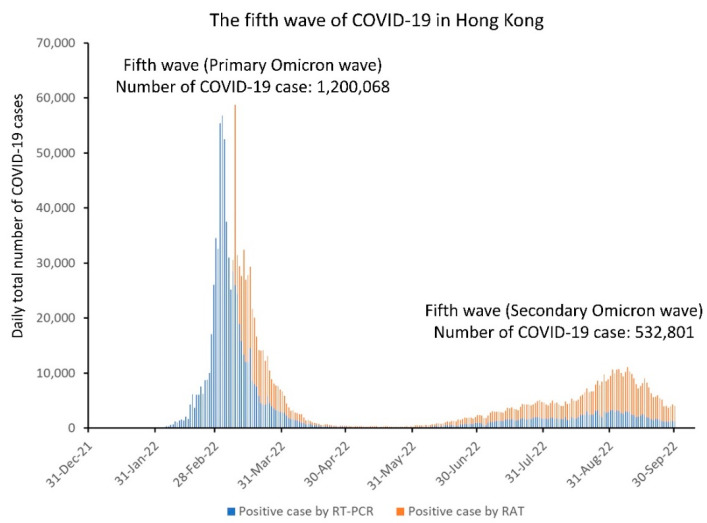
Epidemic curve of the fifth wave of COVID-19 in Hong Kong. RT-PCR, reverse-transcription–real-time polymerase chain reaction; RAT, rapid antigen test.

**Figure 4 viruses-14-02519-f004:**
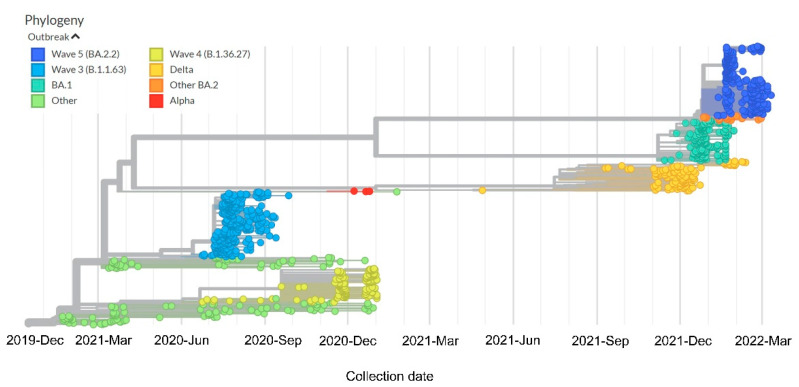
Time-resolved phylogenetic tree of 1168 viral genomes from January 2020 to March 2022 in Hong Kong. The tree was constructed using the Nextstrain command-line interface. The color of the dots represents the outbreak wave of a particular strain. The branch length was determined by the collection date of a sample. Wuhan-Hu-1 was used as the reference sequence during the tree construction.

**Figure 5 viruses-14-02519-f005:**
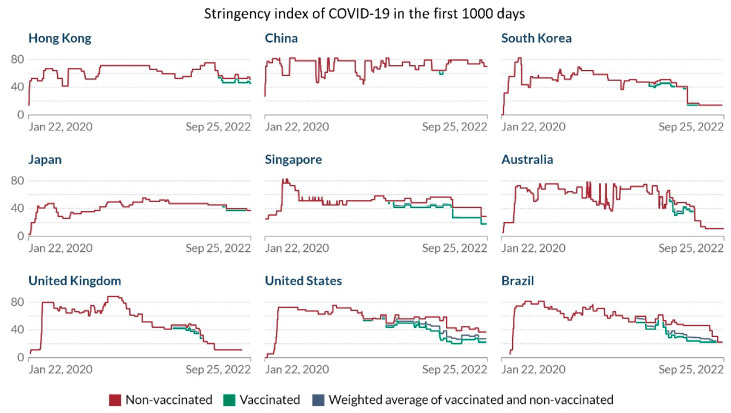
The evolution of the COVID-19 Stringency Index in the first 1000 days. The official announcement of the outbreak of community-acquired pneumonia in Wuhan, Hubei Province, China, was defined as day 1. The figure was generated from the website of the COVID-19 Government Response Tracker [114] and formatted for presentation.

**Figure 6 viruses-14-02519-f006:**
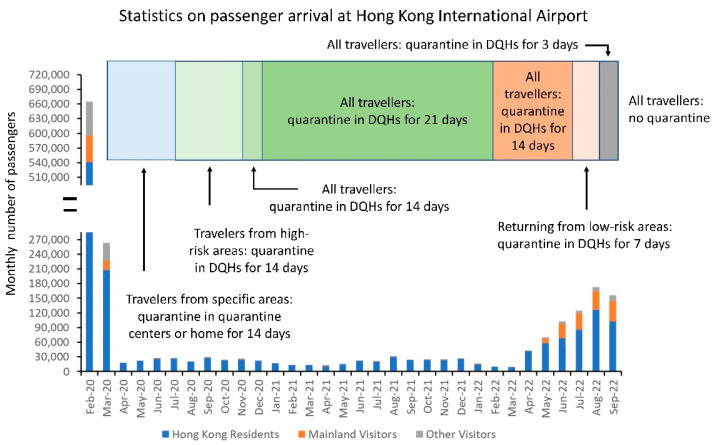
The number of inbound travelers, including Hong Kong residents and visitors from mainland China, arriving at Hong Kong International Airport. DQHs, designated quarantine hotels.

**Figure 7 viruses-14-02519-f007:**
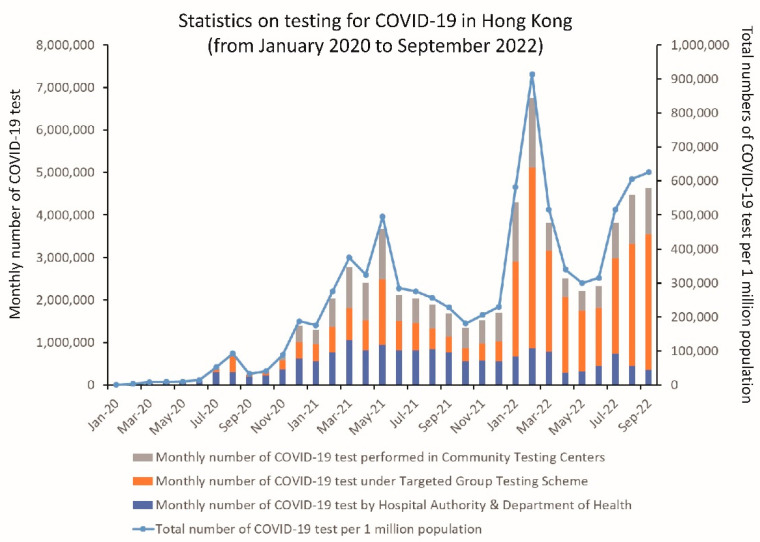
The number of COVID-19 tests under different categories of requests. Community testing centers provided self-paid services (from 15 November 2020, day 321) and free testing for persons under compulsory testing (from 22 November 2020, day 328). The Targeted Group Testing Scheme refers to essential staff providing services to critical infrastructure (from 14 July 2020, day 197). The Hospital Authority and Department of Health were responsible for testing hospitalized patients and out-patients during the COVID-19 pandemic.

**Figure 8 viruses-14-02519-f008:**
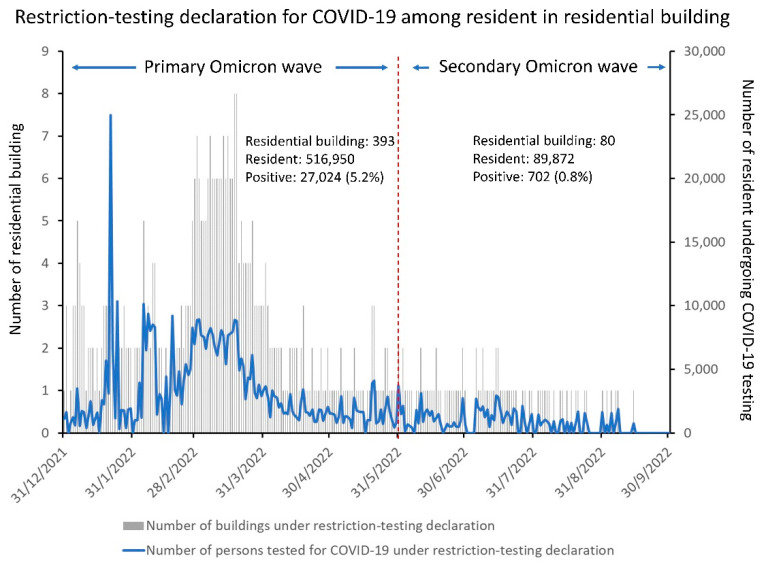
COVID-19 testing in the community under restriction-testing declaration for residents in residential buildings during the fifth wave (primary and secondary Omicron waves) of COVID-19 in Hong Kong.

**Figure 9 viruses-14-02519-f009:**
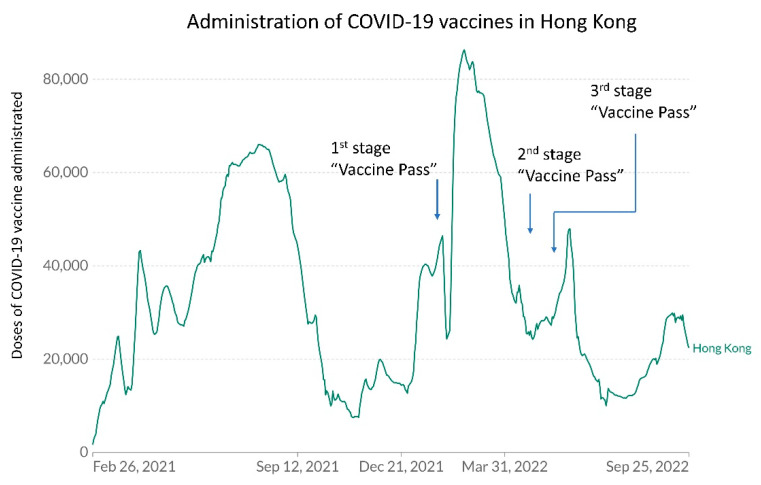
Daily doses of COVID-19 vaccine administrated in Hong Kong from 26 February 2021 (day 424) to 25 September 2022 (day 1000). The figure was generated from the website of “Our World in Data” [140] and formatted for presentation.

**Figure 10 viruses-14-02519-f010:**
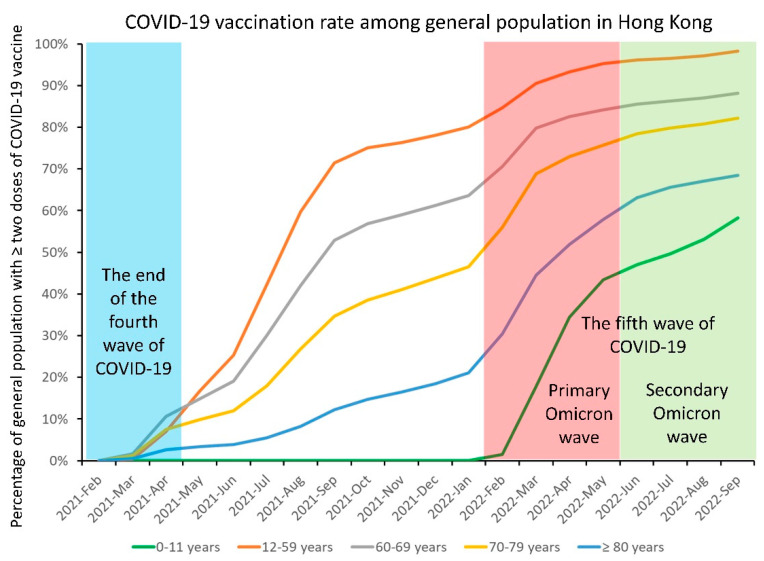
Monthly trend of COVID-19 vaccination rate among the general population until 25 September 2022 (day 1000) in Hong Kong. On 1 August 2022 (day 945), the Scientific Committee on Emerging and Zoonotic Diseases and the Scientific Committee on Vaccine Preventable Diseases of the Centre for Health Protection recommended using CoronaVac for those aged 6 months to under 3 years.

**Table 1 viruses-14-02519-t001:** Global epidemiology of COVID-19 in terms of countries with a cumulative number of confirmed cases of more than 10 million as of 25 September 2022 (day 1000).

Countries	Cumulative Number of Confirmed Cases	Cumulative Number of Deaths among All Confirmed Cases	Incidence per 1000 Population ^a^	Death per 1000 Population ^a^	Case Fatality Rate (%)
United States of America	95,795,378	1,050,631	289.4	3.2	1.1
India	44,560,749	528,487	32.3	0.4	1.2
France	35,056,032	154,854	518.2	2.3	0.4
Brazil	34,624,427	685,750	162.9	3.2	2.0
Germany	32,905,086	149,368	395.7	1.8	0.5
Korea	24,620,128	28,213	480.2	0.6	0.1
United Kingdom	23,621,956	189,919	347.1	2.8	0.8
Italy	22,241,369	176,775	375.5	3.0	0.8
Japan	20,992,896	44,262	166.0	0.4	0.2
Russia	20,694,894	386,551	141.8	2.7	1.9
Turkey	16,852,382	101,068	201.6	1.2	0.6
Spain	13,393,196	113,845	282.6	2.4	0.9
Vietnam	11,467,619	43,146	117.8	0.5	0.4
Australia	10,191,312	14,853	399.7	0.6	0.2

^a^ Population figures were retrieved from the European Centre for Disease Prevention and Control [37].

**Table 2 viruses-14-02519-t002:** Chronology of SARS-CoV-2 variants of concern during the COVID-19 pandemic ^a^.

Nomenclature by the World Health Organization	Earliest Documented Samples (Place and Date)	Duration	Pango Lineage	GISAID Clade	Nextstrain Clade
Alpha	United Kingdom, September 2020	18 December 2020 to 9 March 2022	B.1.1.7	GRY	20I (V1)
Beta	South Africa, May 2020	18 December 2020 to 9 March 2022	B.1.351	GH/501Y.V2	20H (V2)
Gamma	Brazil, November 2020	11 January 2021 to 9 March 2022	P.1	GR/501Y.V3	20J (V3)
Delta	India, October 2020	11 May 2021 to 7 June 2022	B.1.617.2	G/478K.V1	21A, 21I, 21J
Omicron	Multiple countries, November 2021	26 November 2021 onwards	B.1.1.529	GR/484A	21K, 21L, 21M, 22A, 22B, 22C, 22D

^a^ The table was adopted and modified from the World Health Organization [35].

**Table 3 viruses-14-02519-t003:** Epidemiological and virological characteristics in different waves of COVID-19 in Hong Kong.

Wave of COVID-19	Period (Duration of Each Wave, Day) ^a^	Total Number of Cases ^b^ (Death, Case Fatality rate)	Number (%) of Cases in (Episodes of) Community Outbreaks ^c^	Number (%) of Imported Cases (Remark)	Predominant Virus Strain
1st	23 January 2020 (day 24) to 14 March 2020 (day 75) (51 days)	142 (4, 2.8%)	53 (37.3%) (4)	56 (39.4%) (mainly from China)	NA
2nd	15 March 2020 (day 76) to 30 June 2020 (day 183) (108 days)	1064 (4, 0.38%)	130 (12.2%) (3)	739 (69.4%)	NA
3rd	1 July 2020 (day 184) to 31 October 2020 (day 306) (123 days)	4118 (103, 2.5%)	681 (16.5%) (23)	678 (16.5%)	B.1.1.63
4th	1 November 2020 (day 307) to 30 April 2021 (day 487) (181 days)	6451 (101, 1.6%)	1480 (22.9%) (24)	960 (14.9%)	B.1.36.27
Window phase ^d^	1 May 2021 (day 488) to 30 December 2021 (day 731) (244 days)	861 ^e^ (1, 0.12%)	No	854 (99.2%) (without community outbreak	NA
5th (primary Omicron)	31 December 2021 (day 732) to 31 May 2022 (day 883) (152 days)	1,200,068 (9318, 0.78%) ^f^	NA ^g^	2292 (0.19%)	Omicron BA.2
5th (secondary Omicron)	1 June 2022 (day 884) to 25 September 2022 (day 1000) (117 days)	532,801 (585, 0.11%) ^f^	NA ^g^	20,519 (3.9%)	Omicron BA.5

^a^ The waves of COVID-19 were defined according to an epidemiological investigation of community outbreaks or the predominant virus strain; ^b^ From 7 March 2022 (day 798), individuals declaring positive results of rapid antigen detection by self-testing were counted as confirmed cases, in addition to nucleic acid testing by laboratories; ^c^ Outbreaks involving more than 10 cases were reported; ^d^ No COVID-19 outbreaks were reported in the community; ^e^ All COVID-19 cases were diagnosed upon arrival in Hong Kong or during quarantine after arrival in Hong Kong; ^f^ On 23 September 2022, the Centre for Health Protection Department of Health, Hong Kong, announced an additional 153 fatal cases, which were retrospectively reported by the Hospital Authority. They were all fatal cases during the peak months from February to April 2022; ^g^ Due to the huge increase in cases, the epidemiological information of individual outbreak was no longer announced on the website of the Centre for Health Protection, after 6 February 2022 (day 769). NA, not applicable.

**Table 4 viruses-14-02519-t004:** Stepwise enhancement of active surveillance for early isolation of COVID-19 patients in airborne infection isolation rooms (AIIRs).

A	Clinical Criteria (Time of Implementation) ^a^	Remark
1.	Presented with fever and acute respiratory illness or pneumonia (from day 1 to day 23)	Prepare for the importation of index patient to Hong Kong
2.	Presented with fever or acute respiratory illness or pneumonia (with effect from day 24)	In response to the importation of index patient to Hong Kong
**B**	**Epidemiological criteria (time of implementation) ^a^**	
1.	Travel history to Wuhan, Hubei province, People’s Republic of China, within 14 days before onset of symptoms, irrespective of any exposure to wet market or seafood market (from day 1 to day 16)	Prepare for the importation of index patient to Hong Kong
2.	Patient met with one of the following within 14 days prior to the onset of symptoms: (a) had visited Wuhan (regardless of whether the individual had visited wet markets or seafood markets there), (b) had visited a medical hospital in Mainland China, or (c) had had close contact with a confirmed case of the novel coronavirus while that patient was symptomatic (from day 17 to day 20)	In response to the evolving epidemic with increasing number of confirmed cases in Wuhan
3.	Patient met with one of the following within 14 days prior to the onset of symptoms: (a) had visited Hubei Province (regardless of whether the individual had visited wet markets or seafood markets there) or 2(b) or 2(c) listed above (with effect from day 21)	In response to spread of SARS-CoV-2 beyond Wuhan
4.	Universal admission screening for asymptomatic patients (with effect from 9 September 2020 (day 265) onwards)	In response to widespread transmission of SARS-CoV-2 locally

^a^ Active surveillance for patients upon admission according to the clinical and epidemiological criteria was performed from 31 December 2019 (day 1).

**Table 5 viruses-14-02519-t005:** Implementation of the Vaccine Pass to enter specified premises in order to enhance COVID-19 vaccination coverage in Hong Kong up to 25 September 2022 (day 1000) ^a^.

Stage	Time Period	Requirement
First	24 February 2022 (day 422) to 29 April 2022 (day 851)	Persons ≥ 12 years are required to receive at least one dose of COVID-19 vaccine in order to use the Vaccine Pass to enter specified premises.
Second	30 April 2022 (day 852) to 30 May 2022 (day 882)	For persons ≥ 18 years: Two doses of COVID-19 vaccine. For persons aged 12 to 17 years: 1st dose, if within 6 months of 1st dose, or 2nd dose, if after 6 months from 1st dose.
Third	31 May 2022 (day 883) onwards ^b^	For persons ≥ 18 years: 2nd dose, if within 6 months of 2nd dose, or 3rd dose, if after 6 months from 2nd dose. For persons aged 12 to 17 years: 2nd dose, if within 6 months of 2nd dose, or 3rd dose, if after 6 months from 2nd dose.

^a^ Specified premises: all catering business premises (including bars or pubs), amusement game centers, bathhouses, fitness centers, places of amusement, places of public entertainment, party rooms, beauty parlors and massage establishments, clubs or nightclubs, karaoke establishments, mahjong-tin kau premises, club houses, sports premises, swimming pools, cruise ships, event premises, religious premises, barber shops or hair salons, shopping malls, department stores, supermarkets, markets, and hotels or guesthouses. ^b^ The government further extended the Vaccine Pass to cover children aged 5–11 years after 30 September 2022 (1st dose if within 3 months, 2nd dose if after 3 months from 1st dose).

## Data Availability

All the data are publicly available.

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
