# Peer review of "Evolution and Control of COVID-19 Epidemic in Hong Kong"

_viruses, 2022, doi:10.3390/v14112519_

Round 1

Reviewer 1 Report

Solid contribution to science

Author Response

Thank you very much for the comment.

Reviewer 2 Report

This is a well-written and well-structured manuscript. The authors do a good job of summarizing the evolution and control of COVID-19 epidemic in Hong Kong. The factors identified and the analysis performed for the first four waves as well as for the fifth wave are very helpful. The tables and graphics inform the readers greatly. I only have a few minor comments and recommendations.

- Computational simulation plays an important role in mitigating the spread of COVID-19 but is not mentioned in the current manuscript. Representative quantitative modeling outcomes can be included and summarized for assessing the effectiveness of different non-pharmaceutical interventions and vaccination (Section 6).

- Table 1: “%” in the case fatality rate column is missing.

- Figure 2: It is better to use two figures to clearly show the trends of six attributes. In the current version, only three colors (attributes) can be identified.

Author Response

Reply to Reviewer 2

This is a well-written and well-structured manuscript. The authors do a good job of summarizing the evolution and control of COVID-19 epidemic in Hong Kong. The factors identified and the analysis performed for the first four waves as well as for the fifth wave are very helpful. The tables and graphics inform the readers greatly. I only have a few minor comments and recommendations.

- Computational simulation plays an important role in mitigating the spread of COVID-19 but is not mentioned in the current manuscript. Representative quantitative modeling outcomes can be included and summarized for assessing the effectiveness of different non-pharmaceutical interventions and vaccination (Section 6).

Ans: Thank you for the suggestion. We have added the computational simulation in mitigating the spread of COVID-19 in section 6 (6.2 Universal masking and social distancing for COVID-19).

“Multivariate analysis of computational simulation results using the Morris Elementary Effects Method suggests that if a sufficient proportion of the population use surgical masks and follow social distancing regulations, SARS-CoV-2 infections can be controlled without requiring a lockdown.”

Reference:

Li KKF, Jarvis SA, Minhas F. Elementary effects analysis of factors controlling COVID-19 infections in computational simulation reveals the importance of social distancing and mask usage. Comput Biol Med. 2021 Jul;134:104369. doi: 10.1016/j.compbiomed.2021.104369. Epub 2021 Apr 3. PMID: 33915478; PMCID: PMC8019252.

- Table 1: “%” in the case fatality rate column is missing.

Ans: Thank you for the suggestion. We have added “%” in the column of “case fatality rate” in Table 1.

- Figure 2: It is better to use two figures to clearly show the trends of six attributes. In the current version, only three colors (attributes) can be identified.

Ans: Thank you for the suggestion. In Figure 2, the six attributes are “Imported case”, “Epidemiologically linked with imported case”, “Local case”, “Epidemiologically linked with local case”, “Possibly local case” and “Epidemiologically linked with possibly local case”. As the “Imported case”, “Local case”, and “Epidemiologically linked with local case” are predominant, the remaining attributes may not be well visible. Since the other 3 attributes are not constituting a major case load, we propose to keep the present format.

We have made a remark in the footnote of Figure 2.

“Imported cases, local cases, and cases epidemiologically linked with local cases constituted the main burden of COVID-19 cases.”

Reviewer 3 Report

A very comprehensive review and I suggest to accept it in recent form.

Author Response

Thank you very much for the recommendation.
